# Cancer Immunotherapy with Lipid Nanoparticles Loaded with a Stimulator of Interferon Genes Agonist against Renal Tumor Lung Metastasis

**DOI:** 10.3390/pharmaceutics16010031

**Published:** 2023-12-26

**Authors:** Takashi Nakamura, Shun Sasaki, Yusuke Sato, Hideyoshi Harashima

**Affiliations:** Faculty of Pharmaceutical Sciences, Hokkaido University, Kita-12, Nishi-6, Kita-ku, Sapporo 060-0812, Japan

**Keywords:** lipid nanoparticles, STING, renal cell carcinoma, lung metastasis, cancer immunotherapy

## Abstract

Metastatic renal cell carcinoma (RCC) has a poor prognosis, and the major organ of metastasis is the lung. Immunotherapy with immune checkpoint inhibitors (ICIs) is the first-line therapy, but the response rates are low. Thus, the development of a more effective immunotherapy against metastatic RCC would be highly desirable. We previously demonstrated how a stimulator of an interferon gene (STING) agonist-loaded lipid nanoparticles (STING-LNPs) significantly activates natural killer (NK) cells and induces an antitumor effect against cases of melanoma lung metastasis that have shown ICI resistance. In this study, we evaluated the potential of using STING-LNPs in the treatment of lung metastatic RCC (Renca). An intravenous injection of STING-LNPs drastically decreased the amount of Renca tumor colonies. In contrast, monotherapies using ICIs showed no antitumor effect, and even a combination of ICI and STING-LNP therapies failed to enhance the antitumor effects. The main effector cells would be NK cells, and the activation of NK cells by the STING-LNPs may avoid the increased expression of immune checkpoint molecules. These findings provide useful insights into the development of an effective immunotherapy against metastatic RCC.

## 1. Introduction

In the United States, estimates for new cases and deaths from renal cell carcinoma (RCC) in 2022 were 79,000 and 13,920, respectively [1]. The five-year survival rates (2011–2017) for cases of localized and regional RCCs were good (93% and 71%, respectively), whereas the rate for distant RCCs, namely metastatic RCCs (mRCCs), was quite poor (14%) [1]. Since the kidney is a highly vascularized organ, RCC often metastasizes to locations with high blood flow, such as the lungs, bones, the brain, and the liver. The main metastatic destination, however, is the lungs [2].

Employing tyrosine kinase inhibitors (TKIs) and immune checkpoint inhibitors (ICIs) has improved the therapeutic effect for mRCC, and the combinations of ICI/ICI or ICI/TKI and TKI monotherapy are now first-line therapies for the treatment of mRCC. However, a monotherapy with an anti-programmed cell death 1 (PD-1) antibody, nivolumab, showed only a 25% objective response rate [3], and a monotherapy with TKI, sunitinib, reached only 27% [4]. Conversely, the combination therapies increased the objective response rate [4,5]; nivolumab/anti-cytotoxic T-lymphocyte-associated protein 4 (CTLA-4) antibody (ipilimumab) showed a 42% response rate, and nivolumab/cabozantinib (TKI) reached 56%. TKIs inhibit the signaling pathway for vascular endothelial growth factor (VEGF). VEGF enhances the functions of regulatory T (Treg) cells and myeloid-derived suppressor cells (MDSCs), leading to the inhibition of T cell functions and dendritic cell maturation. The administration of TKIs also inhibits Treg cells and MDSCs, which results in the activation of antitumor immunity [6]. Thus, combination therapies using TKIs and ICIs have shown a synergistic effect. These findings suggest that activators of the immune system could be potent candidates for use in combination therapy with ICIs for the treatment of mRCC.

We previously demonstrated that lipid nanoparticles (LNPs) loaded with the stimulator of an interferon gene (STING) agonist, namely STING-LNP, strongly augment antitumor immunity via T cells and natural killer (NK) cells [7,8]. The STING pathway is a critical innate pathway for initiating antitumor immunity [9]. Thus, agonists of the STING pathway are a potent adjuvant for cancer immunotherapy [10]. Additionally, STING-LNPs achieved the induction of therapeutic effects in the case of lung metastasis by a B16-F10 melanoma that was resistant to anti-PD-1 antibody treatment and, with anti-PD-1 antibody, exerted a synergistic effect [11,12]. The intravenously administered STING-LNPs accumulated mainly in the liver and were internalized into the liver macrophages, which led to the activation of macrophages and the production of type I interferon (IFN). The type I IFN activated NK cells throughout the body, including those residing in the lungs. As a result, a synergistic therapeutic effect was generated by the activated NK cells. The synergistic therapeutic effect was dependent on the NK cells. Thus, the administration of STING-LNPs is expected to be effective in treating RCC lung metastasis. However, the effects of STING agonists on RCC do not appear to have been confirmed.

In this present study, we used a Renca lung metastasis mouse model to test the competence of STING-LNPs against mRCC in the lungs. The Renca tumor accurately mimics the growth and metastasis pattern of adult human RCC [13]. We found that the STING-LNPs showed a significant therapeutic effect against Renca lung metastasis and that the effect appears to be due to the activation of NK cells. As far as we could ascertain, this is the first report that describes the potential of this STING agonist as an adjuvant therapy against RCC lung metastasis. In addition, our results may provide new options for strategies that could be used to treat mRCC.

## 2. Methods

### 2.1. Materials and Mice

YSK12-C4 was synthesized as previously described [14]. The commercially available reagents used are as follows: Cholesterol (Sigma-Aldrich, St. Louis, MO, USA); 1,2-Dimirystoyl-sn-glycerol methoxyethyleneglycol 2000 ether (DMG-PEG2k) (NOF Corporation, Tokyo, Japan); Cyclic di GMP (c-di-GMP) (Cayman chemical, Ann Arbor, MI, USA); anti-programmed cell death ligand 1 (anti-PD-L1) antibody (clone: 10F.9G2), anti-CTLA-4 antibody (clone: 9H10), and anti-lymphocyte activation gene 3 (anti-LAG-3) antibody (clone: C9B7W) (Bio X Cell, West Lebanon, NH, USA).

Renca cells were purchased from the American Type Culture Collection (Manassas, VA, USA) and were cultured with RPMI1640 medium (high glucose) containing 10% fetal bovine serum (FBS), 100 units/mL of penicillin/streptomycin (P/S), and 1 mM of sodium pyruvate.

Female BALB/c mice (6–8 weeks old) (Japan SLC Inc., Shizuoka, Japan) were housed under specific pathogen-free (SPF) conditions. The animal experiments described herein were approved by the Animal Committee of Hokkaido University (approval number: 20-0176). All methods were conducted based on the guidelines set by Hokkaido University and the guidelines established for Animal Research: Reporting of In Vivo Experiments (ARRIVE).

### 2.2. Preparation of STING-LNPs

STING-LNPs were prepared as described in our previous reports [11,12]. Briefly, the lipid composition was YSK12-C4/cholesterol/DMG-PEG2k = 85/15/1 (mol ratio). The lipid solution (total 504 nmol) and 500 nmol of c-di-GMP were mixed based on the t-BuOH dilution method. Finally, residual t-BuOH was replaced with PBS using an Amicon Ultra (MWCO 100,000), and unencapsulated c-di-GMP was simultaneously removed. The determination of diameter, polydispersity index (PDI), and zeta-potential was conducted using a ZETASIZER Nano (Malvern Instruments Ltd., Malvern, WR, UK). The amount of c-di-GMP in the LNP solution was determined based on absorbance at 252 nm (e = 24,700). The c-di-GMP recovery rate (%) was determined as follows: 100 × the amount of c-di-GMP contained in the final STING-LNP solution/the amount of c-di-GMP that was first added.

### 2.3. Evaluating the Antitumor Effect against Renca Lung Metastasis

Mice were intravenously inoculated with Renca cells (2 × 10^5^ cells or 4 × 10^5^ cells). The STING-LNP was intravenously injected at a c-di-GMP dose of 4 μg. The ICIs were intraperitoneally injected in the following doses: anti-PD-L1 antibody (200 μg or 300 μg/mouse) [15]; anti-CTLA-4 antibody (200 μg) [16]; and anti-LAG-3 antibody (200 μg) [17]. The dosing schedules for each treatment are indicated in the figures. The mice were then euthanized, and the lungs were collected and measured for weight.

### 2.4. Measurement of mRNA Levels in the Lungs with Renca Metastasis

The intravenous injection of STING-LNPs (4 μg of c-di-GMP) was carried out against mice with Renca lung metastasis on days 2, 6, and 10. On day 11, the RNA was isolated from the collected lungs using RNAiso Plus (Takara Bio, Shiga, Japan) and a Direct-zol RNA MiniPrep (Zymo Research, Irvine, CA, USA) [18]. For RT-qPCR, the primer pairs shown in Appendix A were used. Relative mRNA levels were established using the following equation: (1 + E_R_)^CtR^/(1 + E_T_)^CtT^. The reference gene was β-glucuronidase (Gusb). E_R_ represents the PCR efficiency of the reference gene (Gusb); CtR is the Ct value of the reference gene (Gusb); E_T_ is the PCR efficiency of the target gene; and CtT is the Ct value of the target gene.

### 2.5. Flow Cytometry (FCM) Analysis of Lungs with Renca Metastasis

The intravenous injection of STING-LNPs (4 μg of c-di-GMP) was carried out against mice with Renca lung metastasis on days 2, 6, and 10. On day 12, a single-cell suspension of lung-derived cells was prepared as previously described [11]. Briefly, after the lung tissue was dispersed using a collagenase D treatment, the lymphocyte fraction was obtained using Percoll density centrifugation. Anti-CD16/32 antibody (Biolegend, San Diego, CA, USA) was added to the cell suspension to promote blockage. CD4^+^ T cells (CD3^+^CD4^+^) were identified using an Alexa Fluor (AF) 700 anti-CD3 antibody and an FITC anti-CD4 antibody. CD8^+^ T cells (CD3^+^CD8^+^) were identified with the Alexa Fluor (AF) 700 anti-CD3 antibody and the AF488 anti-CD8 antibody. NK cells (CD3^-^CD49b^+^) were identified using the Alexa Fluor (AF) 700 anti-CD3 antibody and the FITC anti-CD49b antibody. After gating each cell population, the expressions of CD69, CTLA-4, and LAG-3 were determined by using PE anti-CD69, APC anti-CTLA-4, and APC anti-LAG-3 antibodies. All antibodies, including their isotype controls, were products of Biolegend. For the analysis of cells, a CytoFLEX (Beckman Coulter, Indianapolis, IN) was employed. The data were analyzed using FlowJo software (ver. 10.9.0) (BD Biosciences, Franklin Lakes, NJ, USA).

### 2.6. Statistical Analysis

After the normality and the equality of variance were established, either parametric or nonparametric testing was applied. Two comparisons were performed via an unpaired *t*-test (two-tail). Multiple comparisons were performed via one-way ANOVA, followed by either the Tukey–Kramer test (parametric) or the Dunnett’s T3 multiple comparison test (nonparametric). A *p*-value of <0.05 was considered significant.

## 3. Results

### 3.1. Therapeutic Effect against Renca Lung Metastasis via the Administration of STING-LNPs

We first investigated the antitumor effect of the anti-PD-L1 antibody in the Renca lung metastasis model. After the mice were intravenously inoculated with Renca cells, they were intraperitonially injected three times with the anti-PD-L1 antibody (Figure 1A). Numerous tumor colonies were observed in the collected lungs of both groups, and the lung weights were comparable (Figure 1A). Thus, the Renca lung metastasis model represents a resistance model against anti-PD-L1 antibody therapy.

The STING-LNPs used in this study (Figure 1B) were prepared and characterized as reported in a previous study [11,12]. The c-di-GMP was employed as a STING agonist [19]. The characteristics of the STING-LNPs are shown in Figure 1B. Once the free c-di-GMP was removed via ultrafiltration and the zeta-potential of the STING-LNPs became positive, the c-di-GMP was loaded inside the LNP. We had previously reported that LNPs themselves showed neither immune activation nor antitumor effect [8], and the free form of c-di GMP was incapable of functioning as an adjuvant [7,20]. Mice with Renca metastasis were intravenously injected three times with the STING-LNPs (Figure 1C). The STING-LNP treatment drastically decreased the amount of tumor colonies, and the surface appearance of the lungs was similar to that of normal lungs (Figure 1C). The STING-LNP treatment also significantly reduced the lung weight (152 ± 8 mg) compared with that of the PBS-treated groups (350 ± 31 mg) (Figure 1C). The weight of the normal lungs was 127 ± 1.6 mg (n = 6), and the inhibition rate was calculated to be 89%. These results demonstrate that STING-LNP induces a strong therapeutic effect against Renca lung metastasis.

### 3.2. Change in the Immune Status in the Lung with Renca Metastasis

The intravenously administered STING-LNPs accumulated mainly in the liver and were then internalized in liver macrophages, which led to an efficient production of type I IFNs. As a result, the activated NK cells then improved the immune status in the lungs with B16-F10 metastasis [11]. We also investigated changes in the immune status of the lungs with Renca metastasis following treatment with the STING-LNPs. Mice with Renca metastasis were intravenously injected three times with the STING-LNP. The mRNA levels associated with antitumor immunity were measured after the lungs were collected. As a result of treatment with the STING-LNPs, the mRNA levels of *natural killer group 2D* (*Nkg2d*), *Ifng*, *Cd11c*, and *C-C motif chemokine ligand 5* (*Ccl5*) were significantly increased (Figure 2). The increases in *Nkg2d* and *Ifng* were indicative of NK cell activation. A previous report has shown that intratumor NK cells recruited conventional type I dendritic cells (cDC1) to the tumor site via the production of CCL5 [21]. The increases in the mRNA levels of *Cd11c* and *Ccl5* suggested that cDC1 had been recruited by NK cells. These results show that STING-LNPs have the capacity to improve the immune status of lungs with Renca metastasis and that this improvement involves the activation of NK cells.

### 3.3. Effect of Combining ICIs

In general, the change in the immune status in a tumor microenvironment from cold (non-inflamed) to hot (inflamed) improves the response of ICIs [22]. We also previously reported that the STING-LNP/anti-PD-1 antibody combination evoked a synergistic therapeutic effect against B16-F10 lung metastasis [11]. We first investigated the combined effect with an anti-PD-L1 antibody (200 μg/mouse). Unfortunately, therapy combining STING-LNPs and the anti-PD-L1 antibody failed to produce an antitumor effect that was greater than that of monotherapy with STING-LNPs (Figure 3A). Even when we increased the dose of the anti-PD-L1 antibody (300 μg/mouse), no enhancement was observed (Figure 3B). We therefore examined the combination therapy using other ICIs. However, no enhancement was observed in the cases of either the anti-CTLA-4 antibody (Figure 3B) or the anti-LAG-3 antibody (Figure 3C). These results suggest either that only a few immune cells express these immune checkpoint molecules or that other immunosuppressive mechanisms could be involved.

### 3.4. Analysis of Immune Cells in Lungs with Renca Metastasis

Finally, we analyzed the characteristics of CD4^+^ T cells, CD8^+^ T cells, and NK cells in a lung with Renca metastasis. Mice with Renca metastasis were given three intravenous injections of STING-LNPs. On day 12, the lungs were collected, and FCM analysis was performed (Appendix A). The percentage values in Figure 4 represent the percentages for all lymphocytes that were gated in Appendix A. The NK cell population was significantly elevated, whereas the populations of CD4^+^ T cells and CD8^+^ T cells either decreased or remained unchanged (Figure 4A). In addition, most of the NK cells produced CD69, which is an activation marker of immune cells, and this expression in the NK cell population (Figure 4B) indicated that the activated NK cells had migrated to the lung with Renca metastasis.

The expression of immune checkpoint molecules is associated with the activation and exhaustion of immune cells. We also investigated the expressions of CTLA-4 and LAG-3 in CD4^+^ T cells and in CD8^+^ T cells, as well as the expression of NK cells following the treatment with STING-LNPs (Appendix A). In the case of CTLA-4, a significant elevation in CTLA-4 was observed only in CD69^+^ NK cells, but the percentage was relatively low (Figure 4C). In contrast, significant elevations of LAG-3 were observed in all immune cells and in CD69^+^ immune cells (Figure 4D). The fact that the percentages between all immune cells and the CD69^+^ immune cells were similar suggests that an elevated expression of LAG-3 likely occurred in the CD69^+^ immune cells, namely, in activated immune cells. However, the percentage of LAG-3^+^ cells was less than 5% in the case of NK cells, which suggests that the NK cells activated by the STING-LNPs had not suppressed the cancer cells via immune checkpoint molecules in Renca lung metastasis.

## 4. Discussion

In this study, an artificial lung metastasis model was used to demonstrate that STING-LNPs are capable of significantly inhibiting the growth of Renca in the lungs. The STING-LNPs efficiently activated NK cells, and the activated NK cells appeared to be responsible for killing the cancer cells. It is interesting that in the case of Renca lung metastasis, STING-LNPs could have suppressed the expression of immune checkpoint molecules in activated NK cells.

The intravenous injection of the STING-LNPs exerted a strong antimetastasis effect, with an 89% reduction in Renca tumor colonies (Figure 1B). NK cells have a critical role in the elimination of metastatic cancer cells [23,24]. In a lung metastasis mouse model prepared by the administration of B16-F10 cells via the tail vein, NK cells eliminated the injected cancer cells within 24 h after injection [25]. After 24 h, however, the cancer cells degraded the NK cell activating ligands on their surface, which thus avoided further attack by the NK cells and formed metastatic lesions in the lung. At that point, the balance between activation and suppressive signals in NK cells tended to become more suppressive, which resulted in the inability of NK cells to eliminate cancer cells. The STING-LNPs, however, efficiently activated NK cells and increased the activation receptors such as NKG2D on NK cells (Figure 2). Thus, even if Renca cells shed the activation ligands of the NK cells in order to avoid being killed, the signal balance may be prevented from tilting toward the inhibitory mode, and at that point, the activating signal could be maintained in a significant state. In other words, the activation of NK cells by STING-LNPs overcomes the inhibitory signaling from the Renca cells. As a result, the NK cells can kill the Renca cells. In addition, a recent study revealed that in a lung metastasis mouse model prepared by the administration of B16-F10 cells via the tail vein, NK cells, mainly liver NK cells, killed the cancer cells circulating in the blood but not those in the lungs, and the recruitment of T cells into the lungs was enhanced [26]. The intravenously injected STING-LNPs accumulated mainly in the liver and were then internalized to the liver macrophages, where they initiated innate immunity such as the production of type I IFNs that resulted in the activation of systemic NK cells [11]. The findings reported herein indicate that the STING-LNPs appeared to activate the liver NK cells. In the absence of stimulation, such as with adjuvants, however, NK cells cannot eliminate the cancer cells engrafted in the lung tissues [25,26]. In this present study, it is interesting that a significant increase in activated NK cells in the lungs with Renca metastasis was observed following treatment with STING-LNPs (Figure 4A), which indicates that activated NK cells may kill Renca cells. Therefore, we concluded that STING-LNPs exhibit a strong antimetastasis effect against Renca lung metastasis.

It is noteworthy that the use of a combination of ICIs showed no enhancement in the antimetastasis effect of the STING-LNPs (Figure 3). A possible reason for this is that the target molecules of the ICIs used in this study were only involved in the mechanism by which cancer cells escape from NK cells to a minimal extent. Metastatic circulating cancer cells employ various strategies to create conditions in which NK cell inhibitory signals predominate [27]. The first of these strategies is the downregulation of ligands for the activating receptors of NK cells on the surface of cancer cells. The abnormal expression of oncogenic microRNA leads to a decrease in the expression of the MHC class I chain-related protein A (MICA), MICB, and the UL16 binding protein 2 (ULBP2), which are ligands for NKG2D, an activating receptor for NK cells [28,29]. A second strategy is the upregulation of ligands for NK cell inhibitory receptors that are located on the surface of cancer cells. The human leukocyte antigen G (HLA-G) binds to the immunoglobulin-like transcript 2 (ILT2) located on the surface of NK cells, and this suppresses NK cell functions [30]. Although an increase in the level of PD-L1 on cancer cells was reported to inhibit NK cell functions via binding to PD-1 on NK cells [31], its contribution in this study was assumed to be insignificant. In addition to the epigenetic and genetic alterations in cancer cells, transforming growth factor-β (TGF-β) also induces an increase in the level of inhibitory receptors and a decrease in activating receptors on NK cells [27]. In this present study, we used an anti-PD-L1 antibody, an anti-CTLA-4 antibody, and an anti-LAG-3 antibody (Figure 3), but it is possible that these immune checkpoint molecules may not contribute to the escape mechanism of Renca lung metastasis. A third strategy credits the strong therapeutic effect of the STING-LNPs, which is based on the fact that most tumor cells were killed by the monotherapy of STING-LNPs. It is possible that the effect of ICIs could be masked.

In this present study, the activation of NK cells by the STING-LNPs represents a dominant factor in the treatment of Renca lung metastasis (Figure 2 and Figure 4). NK cells have a prominent role in immunosurveillance against metastasis [27,32]. Various studies have demonstrated the antimetastatic effect of NK cells in experimental models [25,33,34,35,36]. In cancer patients, evidence has accumulated regarding the control of metastasis by NK cells [37,38,39,40,41]. The infiltration of effective NK cells into metastatic sites is indicative of a good prognosis. In addition, NK cells circulating in the blood are able to kill the circulating metastatic cancer cells [25]. Thus, treatment with STING-LNPs represents a potent strategy for harnessing NK cells. In contrast, and unfortunately, the combined effects of ICIs were not observed in this study. As mentioned in the previous paragraph, it is possible that Renca lung metastasis may be controlled by other mechanisms that inhibit the action of activated NK cells by STING-LNPs. Antibodies against the ligands for NK cell inhibitory receptors on cancer cells or against TGF-β could be useful for inducing synergistic antimetastatic effects.

In conclusion, the findings reported herein indicate that STING-LNPs induce effective activation of NK cells and exert therapeutic effects against Renca lung metastasis. It is noteworthy that the use of STING-LNPs for this type of treatment could avoid increased levels of immune checkpoint molecules in the Renca lung metastasis model. Since STING-LNPs strengthen NK cells, which are the prominent effector cells against metastasis, the administration of STING-LNPs has the potential to be a potent therapeutic regimen against mRCC.

## Figures and Tables

**Figure 1 pharmaceutics-16-00031-f001:**
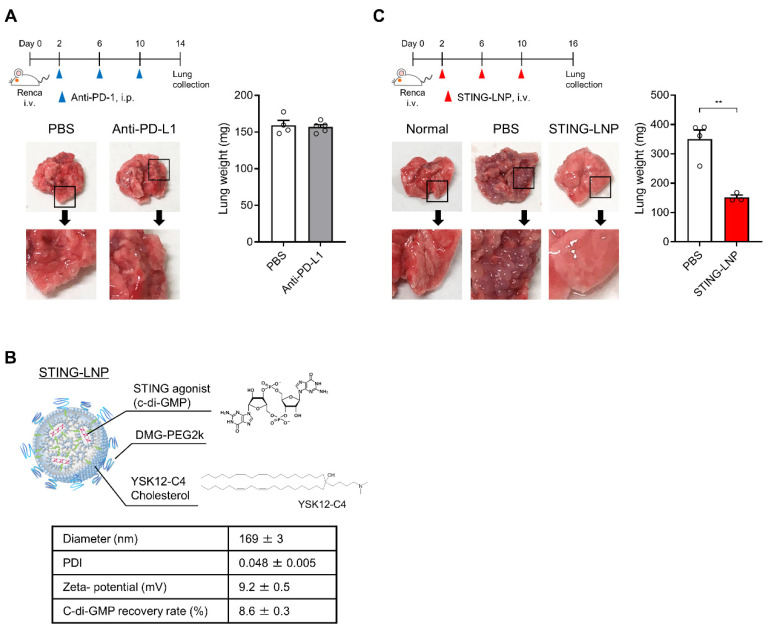
Evaluation of therapeutic effects against Renca lung metastasis in vivo: (**A**) Therapeutic effect of an anti-PD-L1 antibody against Renca lung metastasis. Mice were intravenously inoculated with Renca cells and were then intraperitoneally injected with the anti-PD-L1 antibody (200 μg/mouse) on days 2, 6, and 10. The lungs were collected on day 14. Mean + SEM (n = 4–5). (**B**) Characteristics of STING-LNPs. Mean ± SEM (n = 17). (**C**) Therapeutic effect of STING-LNPs against Renca lung metastasis. Mice were intravenously inoculated with Renca cells and were intravenously injected with STING-LNPs (4 μg of c-di-GMP/mouse) on days 2, 6, and 10. The lungs were collected on day 16. Mean + SEM (n = 3–4, ** *p* < 0.01).

**Figure 2 pharmaceutics-16-00031-f002:**
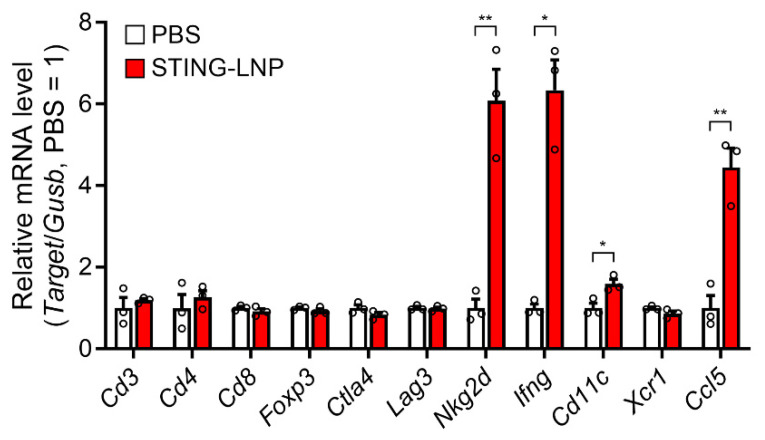
Change in mRNA levels associated with immune responses in a lung with Renca metastasis. Mice with Renca lung metastasis were intravenously injected with STING-LNPs (4 μg of c-di-GMP) on days 2, 6, and 10. On day 11, the lungs were collected, and the RNA levels were measured. Mean + SEM (n = 3, ** *p* < 0.01, * *p* < 0.05).

**Figure 3 pharmaceutics-16-00031-f003:**
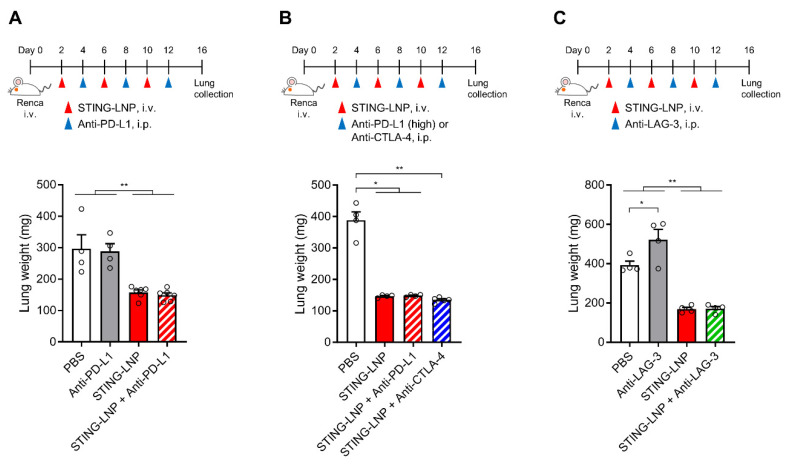
Combination therapy of the STING-LNP and ICIs in vivo. Mice with Renca lung metastasis were injected with STING-LNPs (4 μg of c-di-GMP) and each ICI as shown for each time course: (**A**) Combination therapy with anti-PD-L1 antibody. Mean + SEM (n = 4–6, ** *p* < 0.01). (**B**) Combination therapy with anti-PD-L1 antibody (high dose) or anti-CTLA-4 antibody. Mean + SEM (n = 4–5, ** *p* < 0.01, * *p* < 0.05). (**C**) Combination therapy with anti-LAG-3 antibody. Mean + SEM (n = 4, ** *p* < 0.01, * *p* < 0.05).

**Figure 4 pharmaceutics-16-00031-f004:**
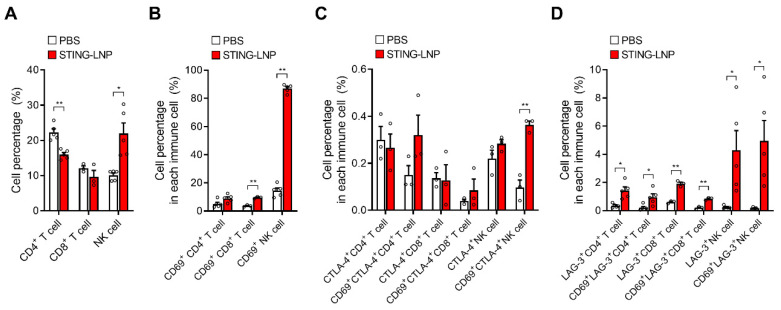
FCM analysis of T cells and NK cells in the lung with Renca metastasis. Mice with Renca lung metastasis were intravenously injected with STING-LNPs (4 μg of c-di-GMP) on days 2, 6, and 10. On day 12, the collected lungs were analyzed. The percentage values represent percentages in all lymphocytes. (**A**) Cell percentages of T cells and NK cells. Mean + SEM (n = 3–5, ** *p* < 0.01, * *p* < 0.05). (**B**) Cell percentages of CD69^+^ T cells and CD69^+^ NK cells. Mean + SEM (n = 3–5, ** *p* < 0.01). (**C**) Cell percentages of CTLA-4^+^ T cells and CTLA-4^+^ NK cells. Mean + SEM (n = 3, ** *p* < 0.01). (**D**) Cell percentages of LAG-3^+^ T cells and LAG-3^+^ NK cells. Mean + SEM (n = 3–5, ** *p* < 0.01, * *p* < 0.05).

## Data Availability

Data are available on reasonable request (to Takashi Nakamura, e-mail: tnakam@pharm.hokudai.ac.jp).

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
