# Peer review of "Cancer Immunotherapy with Lipid Nanoparticles Loaded with a Stimulator of Interferon Genes Agonist against Renal Tumor Lung Metastasis"

_pharmaceutics, 2023, doi:10.3390/pharmaceutics16010031_

Round 1

Reviewer 1 Report

Comments and Suggestions for Authors

The manuscript by Takashi Nakamura, Shun Sasaki, Yusuke Sato, and Hideyoshi Harashima, “Cancer immunotherapy using STING agonist-loaded lipid nanoparticles against renal tumor metastases to the lung,” concern with the possibility of using a stimulator of an interferon gene (STING) agonist loaded lipid nanoparticles in the treatment of lung metastatic renal cell carcinoma (Renca). Intravenous administration of the drug caused effective activation of natural killer cells and had a therapeutic effect against lung metastases. However, no combination effects have been observed with immune checkpoint inhibitors, which are currently used as first-line treatment in these cases. As a result, the authors demonstrated that the developed drug represents an example of an alternative approach to immunotherapy for lung metastatic renal cell cancer, that demonstrating resistance to immunotherapy using immune checkpoint inhibitors. The manuscript can be published in the Pharmaceuticals, but after clarification of some issues indicated below.

1. Although the authors reported that lipid nanoparticles themselves do not exhibit immune activation and antitumor effect in Ref. [8], it would be correct to carry out a control experiment in this work and present the results in Fig. 1B.

2. The authors did not demonstrate the effect of the stimulator of cyclic di-GMP in the experiment described on Fig. 1B. Therefore, it is unclear what the advantages of cyclic di-GMP-loaded lipid nanoparticles synthesized in this work are compared to STIG agonist itself.

3. The authors demonstrated that no synergistic effects were observed when using synthesized lipid nanoparticles and immune checkpoint inhibitors. However, to confirm the lack of therapeutic effect when using anti-CTLA-4 or anti-LAG-3 antibodies, it would be correct to carry out experiments similar to the experiment using the anti-PD-L1 antibody described in Fig. 1A.

Comments on the Quality of English Language

This manuscript requires minor editing of English language.

Author Response

We attached the responses as a file. Please see the attached file.

Reviewer 2 Report

Comments and Suggestions for Authors

1. Please highlight the novelty in introduction.

2. STING-LNP exerted a strong anti-metastasis effect. Could the mechanism be described?

3. Mice with Renca metastasis were given 3 intravenous injections with the STING-LNP. Please explain why only 3 injection was performed. Did the authors try the number more than 3?

Author Response

We uploaded the response file. Please see the attached file.

Reviewer 3 Report

Comments and Suggestions for Authors

Nakamura et al. describe in their work the development of lipid nanoparticles (LNPs) loading a stimulator of an interferon gene (STING) agonist with the objective of activating antitumor immunity to eliminate lung metastases. 

Although this reviewer does not have major considerations from the point of view of the objective, general methodology and results, the way of approaching the work should be improved with some extra studies.

Major comments.

The authors evaluate the activity of STING-LNPs in the remission of lung metastases, which appears to be associated with the activity of local NK cells. However, some issues should be addressed to improve the quality of the manuscript:

1) a more complete evaluation is required to demonstrate the effectiveness of the new formulation. In fact, empty LNPs and free STING molecule should also be included as control groups.

2) the depletion of NK cells would be very interesting to determine the role of these cells in the mechanism of LNPs.

3) a survival curve could be included to demonstrate efficacy of LNPs

4) finally, confocal microscopy showing LNPs colocalization in lung could be very interesting.

Regarding the animal model:

5) can metastases from mRCC behave like B16F10 cells in lung? Or to metastases induced by other cell lines? In other words, is the LNPs sensitivity similar across different cell lines?

6) is Renca or B16F10 cell line innoculated in the in-vivo model? This is not clear.

Figure 3C, anti-LAG3 showed a hyperprogression effect. Perhaps an interesting approach to identify the activity of LNPs could be their administration to this group.

Interestingly, this model can only respond to LNPs, because no effect of ICI antibodies is observed. In this sense, have other dosage regimens assayed?

The authors could consider harmonizing the route of administration, note that i.p. administration may influence the bioavailability of antibodies; furthermore, to improve the robustness of the current results, the authors need to increase the number of animals.

Figure 1 shows a table reporting characteristics of LNPs including the drug recovery rate (%). Is this value releated to the encapsulation efficiency? Is > 90%?

It is important to clarify why the parametric test is used instead of the non-parametric?

Minor comments

The authors should check for several typo errors.

Comments on the Quality of English Language

The authors should check for several typo errors.

Author Response

(The authors gave the same response as above.)

Reviewer 4 Report

Comments and Suggestions for Authors

The manuscript entitled " Cancer immunotherapy with lipid nanoparticles loaded with a STING agonist against renal tumor lung metastasis." by Takashi Nakamura et al. reports the development of a STING-LNP and its potential therapeutic effect against metastatic renal cell carcinoma (mRCC). The study also investigated the potential synergetic effect combining STING and immune checkpoint inhibitor.

Overall, the manuscript is well written and presented. Author applied the previously designed STING-LNP to the mRCC and demonstrated the antitumor effect in vivo. qPCR analysis demonstrate NK cell activation is the key component of this therapeutic effect. The in vivo study combining STING-LNP with ICI, even though didn’t exhibited synergetic therapeutic effect, still provide reference for researchers in this field. However, the manuscript didn’t explain the novelty of this research in comparison to the already published work from same group. Key control is not presented in the result and figures. The manuscript could be considered for publish after addressing the following questions.

1. Figure. 1B, the zeta potential of STING-LNP is positive, which referred by the author demonstrating c-di-GMP was loaded inside LNP. However, in author’s previous study, the Zeta potential is negative. The inconsistence need to be addressed.

“Nakamura, T. et al. Liposomes loaded with a STING pathway ligand, cyclic di-GMP, enhance cancer

immunotherapy against metastatic melanoma. Journal of controlled release : official journal of the

Controlled Release Society 216, 149-157, doi:10.1016/j.jconrel.2015.08.026 (2015).”

If the LNP is positive charge, the nonspecific binding of LNP, especially to endothelial cells need to be investigated as the positive charge LNP could bind to negative charge membrane and trapped one the blood cell membrane.

2. Figure. 3, in this in vivo study, lung weight along was used a measurement of tumor growth, unlike the authors other publication using RLU per whole lung of fluorescent tagged cancer cell.

“Nakamura, T. et al. STING agonist loaded lipid nanoparticles overcome anti-PD-1 resistance in mela-

noma lung metastasis via NK cell activation. Journal for immunotherapy of cancer 9, doi:10.1136/jitc-

2021-002852 (2021).”

To use lung weight as a parameter, whole mouse weight need to be provided as a normalization. Also, a key negative control, which is mouse without Renca inoculation need to be included. Without this control, it is difficult to explain the reason of no difference between STING-LNP and STING-ICI groups. It might be due to the significant tumor inhibition efficacy of STING-LNP that tumor cells are all killed and ICI is not needed, thus no synergetic effect could be observed.

3. The dosage selection of anti-PD-L1 and other ICIs need to be referenced or explained in the manuscript.

4. It would be interesting to see whether the mRCC would be recurred after STING-LNP treatment. Another treatment group with longer observation time could be added.

5. The following paper should be cited

“Woo, Seng-Ryong, et al. "STING-dependent cytosolic DNA sensing mediates innate immune recognition of immunogenic tumors." Immunity 41.5 (2014): 830-842.”

This paper indicate that the host STING pathway is particularly critical for innate immune sensing of immunogenic tumors, a process that results in APC activation, IFN-b production, and priming of CD8+T cells against tumor antigens in vivo.

6. Please add line number in the manuscript for review purpose.

Comments on the Quality of English Language

Overall the manuscript is well written and presented. A few typo need to be corrected

Author Response

(The authors gave the same response as above.)

Round 2

Reviewer 1 Report

Comments and Suggestions for Authors

The authors of the manuscript have done a lot of work to improve it. The manuscript may be published in Pharmaceutics.

Reviewer 3 Report

Comments and Suggestions for Authors

Overall the comments are well addressed.

Reviewer 4 Report

Comments and Suggestions for Authors

I appreciate authors' response of my concerns in the manuscript. All my questions has been addressed. The manuscript could be considered for publication with the current modification.